# Examining charitable giving in real-world online donations

Matthew R. Sisco [1] & Elke U. Weber[2,3]

The current study uses big data to study prosocial behavior by analyzing donations made on the GoFundMe platform. In a dataset of more than $44 million in online donations, we find that 21% were made while opting to be anonymous to the public, with survey results indicating that 11% of these anonymous donations (2.3% of all donations) are not attributable to any egoistic goal. Additionally, we find that donors gave significantly more to recipients who had the same last name as them. We find evidence that men and women donated more when more donors of the opposite sex were visible on the screen at the time of donating. Our results suggest that men and women were both significantly affected by the average donation amounts visible at the time of their decisions, and men were influenced more. We find that women expressed significantly more empathy than men in messages accompanying their donations.

[1] Department of Psychology, Columbia University, 1190 Amsterdam Ave, 318 Schermerhorn Hall, MC5501, New York, NY 10027, USA. [2] Andlinger Center for Energy and the Environment, Woodrow Wilson School, 86 Olden Street, Princeton, NJ 08544, USA. [3] Department of Psychology, Princeton University, Princeton, NJ 08544, USA. Correspondence and requests for materials should be addressed to M.R.S. (email: ms4403@columbia.edu)

Understanding the nature of human altruism has been a core pursuit of science[1,2], examined in extensive laboratory and survey research and theoretical analyses. However, direct observations of non-experimenter solicited charitable contributions have not been analyzed in the peer-reviewed literature to our knowledge. We analyze a unique dataset generated on the online crowdfunding platform GoFundMe and provide novel evidence on several existing hypotheses and lines of research about prosocial behavior.

Our primary data are observations of human behavior in its natural environment. Since these data were produced without an intended analysis, the questions asked of the data were conceived based on the data available. To generate our research questions, we first familiarized ourselves with the publicly available data on the GoFundMe platform, and then surveyed the literature on prosocial behavior to identify hypotheses based on past findings and theories that could be evaluated with the current dataset.

We first briefly describe the dataset to provide context before reviewing the literature and hypotheses we test. The amount donated in each contribution to a GoFundMe campaign is our primary outcome variable of interest. One interesting variable that qualifies each donation is the binary choice of revealing or not revealing one's name to the public, a choice that is telling of one's motivations for giving. Several predictor variables arise from the publicly listed names of donors and recipients. The last names allow for identifying donations made between likely family members when donors and recipients share the same last name. First names can identify the genders of donors and recipients, which opens possibilities for answering questions regarding gender differences in charitable giving and receiving.

At the time of each donation, each campaign page showed potential donors the past ten donation amounts along with the names of the last ten donors and public messages left by them if any. This allows for analyses regarding the influence of the visible presence of the gender composition of past donors, as well as the effects of social comparison on donations. The language in the messages left alongside donations can be analyzed for emotional expressions such as empathy. By analyzing all of the above-mentioned information about each charitable decision, we were able to contribute substantially to several lines of research related to prosociality, described next.

A longstanding question is if prosocial behavior is always ultimately due to some selfish goal. It is important to distinguish between evolutionary and psychological versions of this question[3]. The evolutionary perspective concerns how altruistic dispositions could come to exist in humans. The psychological question asks if it is possible for one person to act with the ultimate goal of increasing another person's welfare[4]. To address the psychological altruism question in the current paper we look for charitable donations made on the platform that we cannot attribute to any egoistic goals as their ultimate motivations[5]. There is no way with our GoFundMe data alone to ascertain whether (even anonymous) donations were ultimately motivated by some egoistic goal. Thus, we estimate the percentage of donations that were not motivated by any egoistic goals using supplementary survey data collected from GoFundMe users. Our first hypothesis is that a significantly non-zero percentage of donations cannot be attributed to being ultimately motivated by any egoistic goals.

The evolutionary question of altruism asks how a disposition to expend resources for the benefit of other individuals could have evolved considering that it would appear to reduce the reproductive fitness of charitable individuals. Three main accounts of how altruism could have evolved in humans are kin selection, reciprocity, and sexual selection[6,7].

Kin selection holds that humans have evolved with a predisposition to take costly actions to benefit genetic relatives[8]. Altruism toward kin is arguably biologically adaptive as such actions would promote continuation of the altruist's genes by increasing the chances of the survival and reproduction of one's relatives. We evaluate the extent to which kin altruism is at play in real charitable donations by quantifying the extent to which donation amounts increase when the recipient is likely related to the donor (i.e., when the donor and recipient share the same last name). Our second hypothesis is that a significantly higher amount on average is donated to recipients who share the same last name as donors than to those recipients who do not.

Another evolutionary explanation of altruism is reciprocity. Trivers[1] posited that sometimes taking a costly action for the benefit of another person can increase reproductive fitness because the person will repay the favor in the future. Computer simulations have demonstrated the plausibility of Trivers' reciprocity theory[9,10] and empirical evidence shows reciprocity at work in the real world[11]. Reciprocity can extend beyond being helped in the future by someone you previously helped. Helping a person can bolster one's reputation for helping and increase chances of being helped in the future by other people. This is known as indirect reciprocity and has been shown to also be a viable account of how altruism may have evolved in humans[12].

A third evolutionary account of altruism is the theory of sexual selection[13]. This theory proposes that acting altruistically can increase an individual's reproductive prospects, in other words increase one's attractiveness to potential mates. Altruism in humans can be thought of in part as an ornamentation purposed to attract the opposite sex[14]. Altruism requires expendable resources and therefore is an honest signal of fitness to potential mates.

Beyond questions surrounding the general nature of altruism in humans, the question of how gender plays a role has intrigued researchers[15,16]. Many studies have shed light on gender differences in altruism, but the sum of past evidence tells a complex and sometimes contradictory story.

The starting point is the seemingly simple question of which gender is more altruistic. Several studies have found that women are more generous than men[17–22] but some report opposite evidence[23,24]. These mixed results suggest that the relationship between gender and altruistic behavior is more nuanced than one gender being categorically more charitable than the other[16].

Indeed, many studies have begun to illuminate numerous motivational and contextual variables that play a role in determining which gender acts more altruistically. Some past work suggests that the gender of the recipient is influential, but findings are mixed regarding which gender tends to receive more generosity[22,23,25,26]. These mixed results make our third hypothesis somewhat open ended. We hypothesized that both male and female donors give different amounts to female recipients than to male recipients.

Related to the theory of sexual selection described earlier, past work[27,28] suggests that people, especially men, may act altruistically in order to emit a costly signal. To test this in the present study we quantified the information that donors had regarding a potential female audience for their generosity by calculating the proportion of female donors visible on the campaign page at the time of each donation decision. Thus our fourth hypothesis predicted that men donate more when more females are present on the page at the times of their donation decisions.

Another context in which men have been posited to be more charitable than women is when they are subject to a social comparison. Meier[24] found that men gave significantly more when told that many others gave than when told that few others did. Women's contributions were not significantly different

across the two conditions. These findings lead to our fifth hypothesis that men are influenced by social comparisons when donating. Operationally, we expect that men give more on average when there are higher visible average contribution amounts from past donors visible on the page when they make their donation decisions.

Finally, another posited factor of importance in charitable decisions is empathy[29]. A meta-analysis by Eisenberg and Lennon[30] suggests that women are more likely to self-report feelings of empathy than men are. Moreover, past work suggests that women are more likely than men to give as the result of empathic concern for recipients[18,21]. These findings lead to our sixth hypothesis that women express empathy more than men in messages accompanying their donations.

The current work analyzes a large-scale dataset of real-world charitable contributions to investigate the abovementioned hypotheses related to human altruism and gender differences in charitable giving.

## Results

**Donations data collection**. We collected records of 558,067 individual donation transactions made to 9,264 campaigns in US dollars. The median donation was $50, and they total $44,249,573 in charitable contributions. The start dates for the campaigns in our dataset range from January 2012 through June 2016.

**Are any donations purely altruistic?**. In total, 21.11% ($SE = 0.06$) of donation transactions in our full dataset were made anonymous to the public. The median anonymous donation was $35. The anonymous donations sum to $10,247,209 which is 23.16% ($SE = 0.3$) of all dollars donated in the full dataset. The anonymous nature of these donations rules out some egoistic motivations such as indirect reciprocity. However we cannot rule out that some egoistic goal was driving these donations such as self-rewards in the form of preserving a positive self-image or direct reciprocity with the recipient (who could see all donors' names). Supplemental survey data is necessary to estimate what percentage of donations were primarily motivated by non-egoistic goals.

To better understand the motivations behind GoFundMe donations we analyzed survey responses from 305 GoFundMe donors. We presented donors with a comprehensive set of potential motivations for making donations on GoFundMe. Respondents reported motivations that influenced their decisions and ranked the motivations from most to least influential. We asked questions designed to probe for every plausible egoistic goal that making a donation may have been seeking to achieve. Figure 1 depicts the proportions of donors that ranked each of the plausible motivations as their primary motivation. Donating primarily because the recipient needed help was reported by 53.4% ($SE = 2.9$) of donors.

This result points to non-egoistic motivations driving the majority of GoFundMe donations. However a more stringent evaluation of Hypothesis 1 is possible with our data by focusing in on anonymous donations. Out of the 305 GoFundMe donors we surveyed, 173 reported they had given an anonymous donation before. We asked each of these anonymous donors the same battery of questions about the motivations behind his or her most recent anonymous donation. We also asked anonymous donors factual questions such as, "Did you ever tell someone about the donation?" in order to provide more opportunities for potential egoistic motivations to be revealed.

In total, 11% (95% $CI = 6.3, 15.7$; $t(172) = 4.6$; $p < 0.0001$; Cohen's $d = 0.35$) of our donors responded negatively to every single question aimed to identify an egoistic goal motivating their anonymous donations. This significantly non-zero amount of donations that are not attributed to any egoistic motivations strongly Hypothesis 1.

**Self-identified donors**. The rest of our hypotheses only pertain to donors who did not choose to be anonymous. After removing anonymous donations, multi-donor donations, and donations that could not be gender estimated, our dataset analyzed in subsequent analyses contains 312,613 transactions made to 8,987 campaigns totaling $21,543,258 in donations. Women made 199,473 contributions (63.81%; $SE = 0.09$) and men made 113,140 contributions (36.19%; $SE = 0.09$; different from 50% at $p < 0.0001$; $t(312,610) = -160.7$; Cohen's $d = -0.29$). In total, women donated $11,928,532 and men donated $9,614,726. Women gave more frequently and more in total, but when males did donate to a campaign, they tended to donate a significantly higher amount ($t(145,290) = -34.9$, $p < 0.0001$; Cohen's $d = 0.14$). The median donation amount was $50 for males and $40 for females.

**Mixed-effects regression**. We implement a mixed-effects regression to evaluate Hypotheses 2 to 5. The outcome variable is the amount of each donation in US dollars. Donor gender is modeled as a dummy variable (donor gender) with 0 representing female and 1 representing male. The results from this regression are displayed in Table 1. We will now describe the remaining predictor variables of interest as they relate to our hypotheses.

**Do donors give more to family members?**. To evaluate Hypothesis 2, that donations are greater when recipients share the same surname as the donor, we created a binary variable (same last name) representing whether the recipient had the same last name as the donor (0 = did not have the same last name; 1 = did have the same last name) and include this as a predictor variable. By using sharing the same last name as a proxy for a family tie we are assuming that when a donor and a recipient have the same last name it is likely that they are related.

We find that when the recipient had the same last name as the donor, the average donation was $29.27 greater ($p < 0.0001$; 95% $CI = 26.38, 32.16$). This effect is statistically highly significant and supports Hypothesis 2.

**Do women receive more than men?**. To evaluate Hypothesis 3 regarding the average amounts donated to male and female recipients, we include gender of recipient (recipient gender; 0 = female, 1 = male) as a predictor and interact this with gender of the donor (donor gender). The estimated coefficient for recipient gender is $-1.15$ ($p = 0.15$; 95% $CI = -2.71, 0.41$) which shows that male recipients received less on average than female recipients when the donor gender was female although this result is not statistically significant. The interaction between recipient gender and donor gender has a coefficient of $-1.06$ ($p = 0.11$; 95% $CI = -2.34, 0.22$) which suggests that men gave even less on average to campaigns created by men compared to campaigns created by women although this finding is also not statistically significant. The direction of these findings is consistent with some past results, however neither of these coefficients are significantly different from zero at the 5% level. Thus, these findings fail to provide support for Hypothesis 3.

**Do men make contributions as a costly signal?**. To test Hypothesis 4, that men give more when more women are visible we model the proportion of female donors visible as a continuous predictor variable (proportion of visible females) ranging from 0

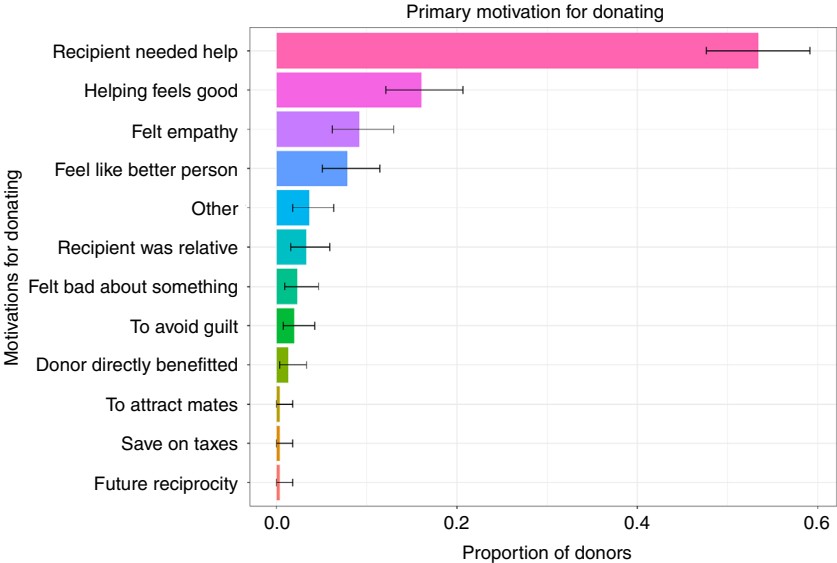

**Fig. 1** Primary motivation for donating. This figure depicts what proportion of donors reported each potential motivation for donating as their primary motivation. Donating primarily because the recipient needed help was reported by 53.4% of survey respondents. Source data are provided as a Source Data file. Error bars depict 95% confidence intervals

| Table 1 Regression results | |
| --- | --- |
| | **DV: Donation amount** |
| Donor gender (male) | 13.89*** |
| | (.83) |
| Proportion of visible females | −2.47** |
| | (.78) |
| Mean visible donation | .05*** |
| | (0.00) |
| Same last name | 29.27*** |
| | (1.47) |
| Recipient gender (male) | −1.15 |
| | (.80) |
| Donor gender (male):Proportion of visible females | 3.29** |
| | (1.15) |
| Donor gender (male):Mean visible donation | .07*** |
| | (.01) |
| Donor gender (male):Recipient gender (male) | −1.06 |
| | (.65) |
| Constant | 58.41*** |
| | (1.85) |
| Observations | 218,053 |

*Notes*: The DV is donation amount is in US dollars. Donor gender is a binary variable where 0 = female, 1 = male. Proportion of visible females ranges from 0–1 representing the proportion of past donors visible on the page that were female. Mean visible donation is the mean centered average donation amount (in US dollars) displayed on the page at the time of each donation decision. Same last name is a binary variable where 0 = donor and recipient did not have the same last name, and 1 = donor and recipient had the same last name. Recipient gender is a binary variable where 0 = female, 1 = male for the gender of the recipient. *P < 0.05; **P < 0.01; ***P < 0.001. SE in parentheses

to 1. This variable quantifies the proportion of visible donors on the page that were female at the time of each donation decision.

If men are motivated to increase their contributions in order to emit a costly signal, we would expect to see higher contribution amounts when a higher proportion of female donors are displayed on the campaign page at the time the donor is making his decision. To investigate this, we model an interaction with donor gender and proportion of visible females to see the effects of visible females on male and female donors.

The significant coefficient of −2.47 (p = 0.002; 95% CI = −4, −0.93) for proportion of visible females suggests that female donors give less as the percentage of visible female donors increases. It was not hypothesized that women would also be affected by the visible presence of the opposite sex, but we do find statistically significant evidence of this. The interaction with proportion of visible females and donor gender has a statistically significant coefficient of 3.29 (p = 0.004; 95% CI = 1.04, 5.54). This suggests that the effect of visible female donors is significantly greater (and positive) for male donors compared to female donors. A regression analysis without recipient-oriented variables, thus with a larger sample size, is shown in Supplementary Tables 5–6 and presents even stronger evidence for this effect including a statistically significant effect of visible females for male donors. These results support Hypothesis 4.

**Are men more influenced by social comparison?**. To test Hypothesis 5, that men are influenced by social comparison, we include the average contribution shown on the page at the time of each donation (mean visible donation) as a predictor. To generate the mean visible donation variable we calculated the mean of the donations visible on each campaign page at the time of each donation decision. We mean centered this variable. We modeled an interaction of this variable with the gender of the donor (donor gender). The estimated coefficient for the mean visible donation is 0.05 and is highly significant (p < 0.0001; 95% CI = 0.05, 0.06) suggesting that the visible previous donations are associated with higher donation amounts of female donors. The interaction term is positive and highly significant with an estimate of 0.07 (p < 0.0001; 95% CI = 0.06, 0.08). This suggests that this pattern also exists for men and is significantly stronger compared to women, which supports Hypothesis 5.

**Do female donors leave more empathic messages?**. Lastly, we test Hypothesis 6 that female donors leave more expressions of empathy in the messages they post with their donations. In total, 84,407 of identified female donors left messages with their donations and 42,232 males left messages. Since the total numbers of messages left by males and females are different in

magnitude, we analyze them for the presence of empathy in a relative manner by calculating the percent of messages expressing empathy left by males or females respectively.

The percent of females who left messages coded as expressing empathy was 12.8% ($SE = 0.12$) and the same percentage for males was 8.3% ($SE = 0.13$). This highly significant difference ($X^2 = 591$; $df = 1$; $N = 126,639$; $p < 0.0001$) supports Hypothesis 6.

## Discussion

The current study analyzed contributions made on the GoFundMe platform to examine psychological hypotheses related to charitable giving. By analyzing hundreds of thousands of real contributions, we provide meaningful evidence contributing to several pre-existing lines of research.

Out of the $44 million in donations, 21% were made anonymously. The fact that these donations were made anonymous to the public marks them as potentially driven by pure altruism, but analyzing supplemental survey data was necessary to shed light on the ultimate goals of GoFundMe donors. We asked GoFundMe donors, i.e., individuals who had previously contributed on the platform, a battery of questions to probe for a broad range of motivations potentially driving their donations. Egoistic motivations included reciprocity, signal-burying[31], self-rewards, avoiding self-punishments, tax incentives, and direct benefits. We sought to include all plausible egoistic goals in this questionnaire but we note as a limitation that other egoistic goals might exist in some donations. Many of these users did respond that one or more egoistic motivations influenced their decisions to help. However, 11% of the users we surveyed responded negatively to every potential self-benefiting motivation for donating anonymously. Thus we suggest conservatively that 11% of the anonymous donations (2.3% of all donations) in our full data set should probably be attributed to purely altruistic motivation.

We note that there may have been many more donations in our dataset that were truly altruistic even though we could not thoroughly rule out egoistic motivations for them. Even many donations made publicly may have been truly altruistic with the purpose of publicly displaying the donors' names being to show support for the recipient rather than for some self-oriented purpose. Out of all the donors we surveyed, 53% reported that their primary motivation was the fact that the recipient needed help.

The main limitation of the survey results is their reliance on self-reports regarding donors' motivations. It is possible that some of our survey participants responded inaccurately in order to conceal their egoistic motivations. Normally, to corroborate a result suggested by surveys and self-reports, experimental methods with consequential outcomes are used to reproduce the finding in observable behavior. In the case of observing pure altruism in humans, such experimental evidence has already been provided[5]. Thus, while our survey data have the normal limitations of reliance on self-reports, their convergence with evidence found in laboratory experiments supports the conclusion that pure altruism can drive human behavior, including charitable donations.

We did not have a hypothesis regarding how the amounts donated anonymously would compare to public donations, but the fact that the median amount donated by anonymous donors was lower than that of public donations is consistent with past literature on observability. A meta-analysis by Bradley et al.[32] found that observability of behavior was associated with an increase in prosociality, with a small but statistically significant correlation across the 117 papers included in the meta-analysis.

Turning to our hypothesis based on the theory of kin selection, we do find that donors gave significantly more to recipients with the same last name. Identifying when donors and recipients had the same last name is a rough proxy for familial ties between them. The effect that we find was likely attenuated by some contributions being made between family members who happened to not share the same last name, and by cases where donors and recipients shared the same last name but actually were not related. The fact that we still see a significant and large effect in light of these attenuating factors speaks to the magnitude of the effect. We note that if donors tended to give more to recipients who shared the same last name but were not family members this would inflate the effect we find, but as described in the methods section we estimate that only a small percentage of same name matches in our dataset were due to chance.

It is interesting to consider what interactions might exist between kin effects and other effects we find such as the effect of visible females or of average visible donations. One might expect donors to react less to normative information and also less to the presence of the opposite sex when donating to kin, because donors' goals are exclusively to help the recipient when they are kin. On the other hand, strong social norms holding that donations to kin should be more generous than to non-kin may make donors especially sensitive to the average amounts donated and to the presence of the opposite sex. As shown in Supplementary Table 8, we ran an exploratory regression model with interaction terms to investigate these questions, and we find that both effects are significantly greater when donating to kin than to non-kin.

Moving to gender differences, our data show that, among participants whose gender could be identified, females contributed $2.3 million more in total than males did in total (out of $21.5 million in total gender-identified donations). It should be kept in mind, however, that 21% of all donation transactions in our data were made anonymously, and the distribution of genders in these anonymous donors is unknown. If substantially more males gave anonymously than females, this could account for the lesser amount donated by males in the non-anonymous donations. It could also be the case that men use the GoFundMe website substantially less than women, and do their charitable giving in other venues instead. Moreover, since it is not possible with our data to determine when multiple donations came from the same person, it could be that individual males donated comparably to individual females in their individual total amounts, but males were more likely to give one large donation and females were more likely to give multiple smaller donations. Given these uncertainties, the aggregate gender differences in our data alone provide limited evidence regarding which gender is acting more generously overall.

Our third hypothesis was that men and women give different amounts on average to male and female recipients. This hypothesis was open-ended regarding the direction of the effect, as past findings were mixed regarding which recipient gender would receive more on average from male or female donors. In our data, both men and women gave more to women than men, but the effects were not statistically significant. The patterns in our findings, while not significant, do line up with the findings of Dufwenber and Muren[22] that female recipients received more donations from male and female donors. Females also received more help than males in an experiment by Colaizzi et al.[25] though Colaizzi et al. did not test an interaction with the gender of the helpers. These results taken together paint a picture that the baseline tendency of male and female donors is to be more charitable to female recipients. However in an experiment by Ben-Ner et al.[23] men gave more to women, and women gave significantly more to men. Thus, more research is needed to ascertain if there is a generalizable pattern of male and female generosity depending on the gender of the recipient.

Past research and theory suggest that men sometimes act altruistically as a costly signal to attract potential mates[27,28,33]. We find support for this in our evaluation of our fourth hypothesis. Men gave more when the visible presence of past female contributors was greater at the time of men's donation decisions. Women similarly gave more on average when the visible presence of past male donors was greater. This result was not hypothesized but is consistent with past research finding that women display more benevolence when induced with mating-related motives[34]. We do not expect that these were conscious decisions to strategically attract mates by donating more when more members of the opposite sex were visible, but rather subconscious inclinations.

Our fifth hypothesis that men are influenced by the visible donations of others was supported in our results, replicating the finding by Meier[24] that males were significantly influenced by social comparison. In monetary terms, our fitted regression model suggests that when the average visible donation was $10 greater than the overall average donation for that campaign, men gave $1.20 more on average while women gave $0.52 more on average with all other predictor variables held constant. Our finding that females may also be affected by social comparison (though less so) was not seen in the results of Meier[24]. Since we find that the association is about twice as strong for men as it is for women, it could be that the experiment of Meier[24] was simply underpowered to detect the effect for female participants. These findings taken together suggest that charitable organizations may benefit from displaying large recent donations, with men likely responding more to the information than women.

Lastly, we also found support for our sixth hypothesis based on past findings that women may be more motivated than men to make charitable contributions due to empathic concern for recipients[18]. We found that women were more likely than men to express empathy in messages accompanying their donations. We note that while these findings are consistent with the expectation that women are more motivated by empathy, we cannot rule out the possibility that this effect may be due to women simply expressing empathy more than men and not necessarily being motivated more by it. There is no past literature to our knowledge suggesting that men are less likely to express empathy given that they are feeling it, but we acknowledge this as a possibility.

Beyond the evidential contributions this study makes to important lines of research on prosocial behavior, the current paper aims to set a pioneering example of leveraging naturally generated data to further scientific inquiry. At present, digital traces of real-world consequential human decisions are increasingly generated by thousands of digital platforms of which GoFundMe is one of. Harnessing such naturally generated data to contribute to scientific research as we have done in the current study can fill in gaps left by the limitations of traditional methods such as surveys and experiments. Of course, observational data such those analyzed in this paper have their own limitations, in part because they are not generated with specific hypotheses or analyses in mind. Analyzing such data requires some departures from traditional procedures, such as incorporating natural language processing techniques and formulating questions based on the available data rather than designing data collection procedures to answer predefined questions. Importantly, we note that since our data are observational (rather than experimental) the causal implications of our results must be interpreted with this limitation in mind. We hope that this paper provides a useful example for other researchers to effectively seek out and analyze large-scale, naturally occurring datasets of consequential human behavior to contribute to behavioral science.

To summarize, using a massive dataset of donations made on the crowdfunding platform GoFundMe, we contribute valuable evidence to an ensemble of questions addressed in past work related to human altruism and the influences of gender on charitable giving. With this novel dataset, we successfully replicated past findings that gender, familial ties, an inclination to signal the possession of resources, social comparison, and empathy all play roles in charitable giving. Based on supplemental survey data we estimate that 11% of anonymous GoFundMe donations (translating to 2.3% of all donations) in our dataset cannot be attributed to egoistic motivations and thus represent pure altruism. We note that our sample is composed of primarily western donors, and thus our findings may not be generalizable to other cultures. Future research on this topic should further explore the cross-cultural variability of altruistic tendencies[35–37]. By better understanding human altruism and the relationship between gender and charitable giving, society can improve its efforts to increase prosocial behavior in humans.

## Methods

**Donations data collection and processing.** The GoFundMe platform is a popular website launched in 2010 which allows anyone to create campaign pages describing requests for charitable financial donations. The types of campaigns on GoFundMe include charitable causes such as disaster relief funds and assistance for medical bills. All categories of campaigns in our data can be seen in Supplementary Fig. 2 along with the distribution of male and female contributions to each campaign category. Medical campaigns are the most common type in our dataset. People could voluntarily browse these pages and make donations to requesters. Records of years of donations were publicly displayed on the GoFundMe website at the time of this writing. Donors could (and often did) leave public messages along with their donations.

When donors were browsing a requester's page, they saw on that page the requester's goal amount, the amount of money donated so far toward that goal, a picture posted by the requester, a description of the request posted by the requester, and the names, donation amounts, and messages of the last (up to) 10 people to make a donation to that campaign. An example of a requester's campaign page is shown in Supplementary Fig. 1.

The data were downloaded from the GoFundMe website during May and June of 2016. Information from the campaign pages as well as all donation amounts to each campaign were downloaded from the publicly accessible GoFundMe website using the statistical software R.

In the self-identified donors analyses (hypotheses 2 through 6) we removed anonymous donations and also donations with names that described more than one person (e.g. "John and Mary Smith" or "Alice & Deborah") as our hypotheses focus on the gender of a single donor behind each transaction, not a group. In sum, 5.3% of donations were removed for indicating multiple donors. The donation transactions for which the gender of the donor could not be confidently estimated were also not included in these analyses.

**Gender estimation.** The gender of each requester and donor was estimated using information about their first names from the U.S. Census. We used the name frequency data from the 1990 US Census. The U.S. Census program provides the percentages of people with commonly occurring first names that are male or female. Thus, for any relatively common first name the census data provide an empirical probability that the person is male or female given his or her name. We used these probabilities to estimate the genders of requesters and donors by comparing the probability that a person with that first name is female to the probability that a person with that name is male. Our conservative and effective gender estimation algorithm is provided in Supplementary Methods 2.1. We only labeled a participant as male or female if he or she was above 10 times more likely to be one gender than the other given the gender frequencies associated with his or her first name. This gender estimation procedure confidently estimated 76% of the participants in our database who provided their names (and did not indicate multiple donors). Supplementary Table 1 shows a random sample of the gender detection output given user-provided names.

We coded the genders of the campaign creators with the same algorithm used for coding the names of donors. Out of the original set of campaigns, 6,100 campaigns were successfully coded for the gender of their creators and are included in the main regression analysis.

**Survey data collection.** We recruited 331 participants through Amazon Mechanical Turk. Participants were compensated at the equivalent of $15 per hour. In the summary of the survey we explained that it was only for people who previously donated on GoFundMe. We also asked at the beginning of the survey if they had donated to a campaign on GoFundMe before and disallowed participants who answered "No" from continuing with the study. Prior to analysis we removed 16 participants for failing an attention check question. We reviewed the open ended responses to identify unacceptable responses and removed ten participants

for leaving illegitimate (clearly automated or incomprehensible) responses or ones that indicated they donated on a different crowdfunding platform than GoFundMe. This left a sample of 305 responses. 47% of participants were male and the sample had a mean age of 33 ($SD = 11$). The modal interval of amounts donated was $16-$30. In collecting our human subjects data we complied with all relevant ethical regulations for work with human participants. Informed consent was obtained from participants. This study was approved by the Princeton IRB under protocol #11464.

We asked all participants a battery of questions regarding their motivations for making their most recent donation. These questions were designed to address all potential egoistic motivations including seeking self-rewards (e.g. warm glow or positive view of self), avoiding self-punishment (e.g. guilt), seeking social rewards, avoiding social punishments, costly signaling, tax incentives, reciprocity, and directly benefitting from the campaign. Respondents that gave a donation anonymously then proceeded to answer the same set of questions and others regarding their most recent anonymous donation. More details on this survey can be found in Supplementary Methods 2.11.

**Statistical framework**. We implement a mixed-effects regression to evaluate hypotheses 2 to 5. The outcome variable is the amount of each donation in US dollars. Donor gender is modeled as a dummy variable (donor gender) with 0 representing female and 1 representing male.

There were some extreme outliers in our data due to a small number of donors giving massive amounts, so we excluded observations where the amount donated was greater than or equal to three standard deviations from the average donation amount or where the mean visible donation on the page was three standard deviations above the average mean visible donation on the page at the time of donating. When we run the same regression but with no outliers excluded, the pattern of significant findings is essentially the same and the coefficients all increase in magnitude. This can be seen in Supplementary Table 7. As a robustness check, we ran the same regression analysis with cutoff thresholds of 2 and 1 standard deviations. The effects are virtually the same under these different cutoff levels and can be seen in Supplementary Table 7. Statistical comparison of mean amounts of donations by men and women was performed using a two sample (two-tailed) t-test allowing for unequal variances across groups. The mean donation for males was $84.98 and for females was $59.80.

In our regression analysis, the campaign each donation was made to and the category of each campaign were both modeled as random intercepts. Including effects for campaigns in the model addresses the fact that different campaigns inherently have different baseline amounts that are appropriate for donations. For example, donating to a couple's honeymoon fund likely does not warrant the same donation amount as donating to a fund for an urgently needed surgery. Modeling a random intercept for each campaign controls for potential campaign-level confounds such as the popularity of causes or the socioeconomic status of geographic areas where campaigns originated. Similarly including effects for campaign categories is to account for baseline differences across categories.

We note that by including the recipient-oriented variables in the same regression as the donor-oriented variables, we reduce the sample size notably. We include only the full model regression results in the paper for simplicity, as the results even with a smaller sample size are almost identical to the results with the recipient-oriented variables excluded. Supplementary Table 5 provides the regression output with the recipient-oriented variables excluded.

**Same name analyses**. Regarding Hypothesis 2, we note that only about 1% of donations were made to recipients with the same publicly listed last names as the donors. This is a small percentage, but still left >1400 transactions made to an apparent relative for each gender. We do not know how many donor-recipient pairs were relatives that we could not identify as relatives, so our data cannot be used for inference regarding the frequency of donations to relatives versus non-relatives.

By using sharing the same last name as a proxy for a family tie we are assuming that when a donor and a recipient have the same last name it is likely that they are related. In order to get a sense of the likelihood of donors and recipients sharing the same last name due to chance and not familial ties, we evaluated the empirical probability of this using our own dataset. We find that around 2% of cases where donors and recipients share the same last name in our dataset are likely due to chance rather than familial ties. In Supplementary Methods 2.6 we provide a more detailed account of this robustness check. Also in Supplementary Methods 2.6 we provide calculations demonstrating that with a small probability of random last name matches between donors and recipients (on the order of 2%) it is unlikely that random matches are affecting our results in a substantial way.

Instances where a person donated to a campaign of someone with the same last name but who was not a family member could add noise to the analysis and attenuate the effect size but should not increase the probability of a type I error, assuming there is not a substantial effect on donations between people who share the same last name by chance. If there was such an effect, this would affect our results subtly but not substantially due to the low probability we found chance matches were likely to have occurred with as described above. Instances where a donor gave to a recipient whom is a family member but did not share the same last

name (e.g., donating to a sibling who married and took her spouse's last name) will attenuate the estimated effect, but would not result in a type I error.

Men and women have different probabilities of sharing last names with family members due to conventions of name-taking in marriages. Therefore we do not attempt to ascertain gender differences in kin generosity with this variable.

**Proportion of visible females**. Testing our fourth hypothesis involved calculating the proportion of visible females. On a GoFundMe campaign page a prospective donor could see the names and donation amounts of the last 10 people to donate to the campaign. Since we know the order of the donations, we are able to determine which past donors and donations were shown on the page at the time of each contribution. If one of the past donations seen on the page was anonymous, could not be gender-coded, or was from more than one person then the donor was not included in the calculation of the female proportion of visible donors. In other words, the proportion of visible females variable represents the proportion of gender-identifiable donors visible on the page that were female. If all visible donors on the page were female then proportion of visible females would be equal to 1. If half of the visible donors were female it would be equal to 0.5.

Since donor gender is modeled as a dummy variable (0 = female, 1 = male) and there is an interaction in the model with donor gender and proportion of visible females, the coefficient for proportion of visible females can be interpreted as the effect of the proportion of visible females for female donors. Donor gender would be equal to 0 for females, so the Donor gender:Proportion of visible females coefficient is multiplied by zero and removed from the equation.

We note that the costly signaling effect for female donors did not remain significant in one robustness check where we excluded outliers above one standard deviation from the mean donation and mean visible amount. These results can be seen in Supplementary Table 7.

**Mean visible donations**. Since females tend to give less per donation, the variable mean visible donation is correlated with the variable proportion of visible females. The more females on the page, the lower the average donation shown tends to be. By modeling both of these variables simultaneously in one regression model, we avoid the potential issue that only one of these effects is truly at work here since they both explain unique portions of the variance (i.e. both have significant effects).

It is plausible that time could be a time confound in the relationship between the mean visible donations and the amounts given by each donor. That is, if there are consolidated periods in time when donations increase across donors and campaigns, such as on a holiday that encourages generosity, this may act as a third variable and increase the average visible donations and the individual donations simultaneously. This would give the impression of donations being influenced by prior donations because they would be correlated. We investigated the possibility of this by looking at the correlations between each donation and the past 20 donations. Since only the past 10 were shown on the screen, if the effect we see is due to visibility then there should be a stark drop-off in correlations after the tenth. This is the pattern we find which can be seen in Supplementary Figs. 3, 4.

**Empathy coding**. In order to assess the content of messages for expressions of empathy, we developed a short list of empathic phrases such as "I empathize…", "… feel your…", and "… heartfelt…". The full algorithm including all key phrases can be seen in Supplementary Methods 2.5. We automatically coded for the presence of these phrases in the messages left by donors as a binary indication of each message expressing empathy or not. We implemented a permutation-based robustness check to ensure the validity of this method which can be seen in more detail in Supplementary Methods 2.5. Standard errors for percent of transactions expressing empathy were calculated using the normal approximation for binomial standard errors.

**Reporting summary**. Further information on research design is available in the Nature Research Reporting Summary linked to this article.

## Data availability

The data that support the findings of this study are available via a public repository accessible at https://doi.org/10.7916/d8-cckc-3f61. All of the code needed to replicate the analyses presented is included in the main Supplementary Information document. A reporting summary for this article is available as a Supplementary Information file. The source data underlying Fig. 1 are provided as a source data file.

## Code availability

All code necessary to perform main and supplementary analyses is included in the supplementary materials. This code was originally written and implemented in R version 3.5.1.

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

## Acknowledgements

This research was funded under the cooperative agreement NSF SES-1463122 awarded to the Center for Research on Environmental Decisions and also by the National Science Foundation through the award NSF-1144854 "IGERT: From Data to Solutions: A New PhD Program in Transformational Data & Information Sciences Research and Innovation".

## Author contributions

M.R.S. compiled the dataset and performed the analyses. M.R.S. and E.U.W. together designed and interpreted the final analyses. M.R.S. wrote the initial draft of the manuscript. M.R.S. and E.U.W. worked together on revising and further developing the manuscript.

## Additional information

**Competing interests:** The authors declare no competing interests.

