## [Peer Review File · Nature Communications]

Reviewers' Comments:

Reviewer #1:

Remarks to the Author:

Review of Sisco and Weber Charitable Giving

This manuscript reports on a big data study of online charitable donations made via the GoFundMe platform. The total sample of donations was more than half a million and the total amount donated over \$40 million. For some comparisons only a rather smaller subsample of the data was usable, but the sample size was still quite large relative to purposed collected experimental samples.

The study is limited by the nature of the online data retained by GoFundMe. For example, demographic data on donators and recipients is limited to first and last names (unless donations were anonymous). The authors utilized first names to characterize gender and last names to check for kin based giving. Their procedures for using names in lieu of more direct demographic data are sensible and they are candid about their resulting imperfections. They make a strong case for observational studies, experiments, and questionnaires as having complementary methodological strengths and weaknesses. When the findings of multiple methods are consilient we can be rather more confident of them than if we are reliant on only one.

The headline result of this study is that 21% of GoFundMe are made anonymously. This is significant because there is a large controversy in the literature over whether all cooperative behavior can be cashed out in the end in terms of individual costs and benefits or whether there is a significant amount of genuine selfless altruism in human behavior. The authors take their results to be robust support for the later alternative. I think they are correct.

The authors also find support for kin altruism being important, for a sexual selection hypothesis, and for a social influence effect. They do not find support for the idea that women give more than men but women do express emotional motivations for giving more than men. These findings are all consistent with previous reports from experimental and questionnaire based studies and with theoretical expectations. Thus this study does find consilient support for several important hypotheses.

The theoretical discussion (lines 155-294) leading to the 6 hypotheses the authors' test is not as strong as it should be. The authors don't seem to be aware that the evolutionary theory of altruism has progressed far beyond Trivers and Hamilton, as important as these classical insights are. Steward-Williams (2015) is a decent review.

The authors might want to briefly situate their findings in the larger literature on variation in prosocial behavior. The very large experimental literature on economic games is interesting in that in such games as the Public Goods Game with punishment, cooperation in many samples is nearly perfect after ten rounds if participants are given tools like punishment or cheap talk to encourage cooperation (Baum et al. 2012). In unpublished work in my lab it seems like on the order of 20-30% of participants act as altruistic leaders in establishing cooperation. They cooperate themselves, use whatever tools like punishment and persuasion they are given to encourage a reluctant majority to cooperate. I think other peoples' experience is similar. It is also worth noting that prosociality is highly variable cross-culturally (Henrich et al. 2010). For example, Herrmann et al. (2008) using the PGG with punishment discovered a number of societies in which anti-social punishment was common. In these groups cooperation rates remained low. Both the Henrich and Herrmann used observational data to argue that their findings had external validity. The substantial minority of altruists Sisco and Weber found seem to be important out of proportion to their numbers.

Stewart-Williams, Steve

2015 Morality: Evolution of. In International Encyclopedia of the Social & Behavioral Sciences edited by J. D. Wright, pp. 811-818. 2nd ed. vol. 15. Elsevier, Oxford.

Baum, William M., Brian Paciotti, Peter Richerson, Mark Lubell and Richard McElreath

2012 Cooperation due to cultural norms, not individual reputation. Behavioural Processes 91(1):90-93.

Herrmann, Benedikt, Christian Thöni and Simon Gächter

2008 Antisocial punishment across societies. Science 319:1362-1367.

Henrich, Joseph, Robert Boyd, Samuel Bowles, Colin Camerer, Ernst Fehr and Herbert Gintis

2004 Foundations of Human Sociality: Economic Experiments and Ethnographic Evidence from Fifteen Small-Scale Societies. xix + 451 ed. Oxford University Press, Oxford.

Reviewer #2:

Remarks to the Author:

Nature Comms Report

I apologize about the delay in providing this report. I had several other reports that I needed to complete first and since I am also teaching this semester, it took me a while to do so. I know such a delay can be frustrating for the authors. Thanks for your understanding.

The authors analyze a large dataset of GoFundMe donations. They find that some 20% of donations are anonymous, that donors give more to family members, that donors give 5c more for every additional dollars others have given. They also find interesting gender differences: men respond more to the amount others have given, and give more when a higher proportion of other donors are female. People's justification of why they give also varies by gender.

This paper makes a substantial contribution by analyzing such an extensive dataset of real-world contributions. The findings meaningfully help to advance our understanding of altruism. Indeed, I believe that with some revision, this can be a first-order contribution. However: (1) I strongly disagree with the current framing. I will try to explain why and will suggest an alternate framing. (2) The paper would benefit from extending the analysis of donations to kin vs. non-kin, and focusing more on this.

First, my main criticisms regarding the framing:

- I would cite the literatures on norm enforcement and indirect reciprocity. I like to cite Boyd (Ch. 2 of "A Different Kind of Animal" for instance) on norm enforcement. I typically cite Nowak's "five mechanisms" on indirect reciprocity. But there are lots of options.

- The argument confuses "proximate" (what we think and feel) and "ultimate" explanations (what shaped those ideologies and preferences). For instance, consider this sentence, "Do humans take costly actions to help others for exclusively egoistic reasons, or are other-regarding goals also at play?" That's a false dichotomy. We typically think of "other-regarding goals" as likely having developed for reasons that, ultimately (though not necessarily consciously) benefit the individual. That is the contention inherent in the norm enforcement and indirect reciprocity literatures. Of course, that doesn't mean we consciously deliberate over the egoistic benefits of donating. Partially this is because, in general, we often aren't aware of the motivations that drive our ideologies and preferences. We historically had no idea that cheesecake tasted great because it was a calorie bomb—

we just found sweet and fatty foods tasty. We don't typically deliberate over the fertility of a potential mate, we just find them hot. Etc. And, partially, it's because in the domain of altruism, people benefit from being seen as someone who doesn't deliberate over the benefits to themselves. For more details see <http://www.pnas.org/content/112/6/1727>. For instance: "First, psychologists and philosophers have long asked the following question: is helping others "always and exclusively motivated by the prospect of some benefit for ourselves, however subtle" (20) [for example, the conscious anticipation of feeling good (21), avoidance of guilt (22–24), reputational benefits, or reciprocity (1–14)]. At the extreme, this question amounts to asking if saintly individuals, such as Gandhi or Mother Teresa, were motivated thus or if they were authentic altruists who did good without anticipating any reward and would be altruistic, even in the absence of such rewards. Our model suggests that authentic altruism is, indeed, possible: by focusing entirely on the benefits to others, authentic altruists are trusted more, and the benefits from this trust outweigh the risk of, for example, dying a martyr's death." (To be clear, I don't think you need to cite this paper. I just hope it helps to clarify the issues raised by your existing framing.)

Here is how I would, instead, frame your contribution. (Notice the more central role of the kin vs. non-kin contrast.)

- A large literature identifies reputations as a major driver of altruism towards non-kin. Cite the literatures on norm enforcement and indirect reciprocity. Maybe also cite the costly signaling paper.
- Emphasize that this is not necessarily conscious.
- This is thought to explain many otherwise puzzling features of altruism, like why people care so much if they are being observed (cite the observability lit), why they don't attend so much to effectiveness (cite the scope insensitivity literature), and why they attend so closely to normative information.
- This paper is the first to provide an analysis of a substantial number / \$\$ value of real-world donations and show that the patterns of giving are consistent with these predictions of these literatures.
- A unique feature of this dataset that makes it especially useful for this purpose is the the ability to observe and contrast the donation behavior of kin and non-kin. Critically, since altruism to kin is driven not by reputations, but by kin selection, the puzzling features discussed above are expected to be weaker for kin.
- The key findings are:
 - The vast majority of people don't give anonymously. When they do give anonymously, they give less
 - People give substantially to kin, both in terms of number of donations and their avg size.
 - Ideally, here you would discuss that people aren't so sensitive to recipient's need (see below); but are much more sensitive when giving to kin.
 - People respond substantially to normative information about what others give. But, less so when giving to kin.
- There are even some gender differences that conform to this hypothesis:
 - Men give more
 - Men are more "competitive" in their giving, giving more when more others give
 - Men give more when more females observe the gifts
 - Ideally, you'd show these effects go away for kin
 - In addition, there are some interesting gender differences in how people justify donations

I suggest the following additional analyses that would support this framing, and, IMO, generally improve the paper:

- Can you please run an interaction between `same_last_name` and `mean_visible_donation`? You should find that the effect of `mean_visible_donation` is smaller for kin.
- Similarly, `prop_visible_female` should matter less for kin. (Basically, I would interact kin with everything, and showing that it always reduces your coefficient.)
- Can you show that people are generally not-so-responsive to need somehow? I.e. show that they give when lots of others are giving, even if the campaign doesn't need it? And... can you then show that kin tend to be more sensitive to the recipient's need than non-kin?
- Can you tell what proportion of a given recipient's gifts came from kin?
- Ideally, you'd be able to say more about what's different about anonymous donations. Do you have the data to analyze who the donors were? Or is all you can see is a line of data with an anonymous donation?

Here are some additional suggestions and comments:

- I had a bit of trouble telling what your data look like. Perhaps you can clarify this further? (In particular, as you can tell from the above, I would like to see some analyses at the recipient level, but I can't tell if you can do this using your data and are constrained to the particular variables in Tbl. 1. Another question: Is the only information donors can see the "visible" donations, or can they dig around and get more info? Is that the only info you can see?)
- A glaring omitted cite is to Croson and Gneezy's review of gender differences in charity. I advise familiarizing yourself with this paper and the literature in it, and framing your contribution on gender differences accordingly. This comment is intended to be constructive, as I think you add a lot by showing that the magnitude of gender differences is large in this important, real-world setting.
- Tbl. 1 is woefully inconsistent with the norms of general science journals and should be reformatted. E.g., use English variable names, and add a figure legend that allows the table to be interpreted on its own. What is the interpretation of the coefficients? What units are the variables in? Etc.
- IMO, you're not making enough of the setting. There are so few analyses of real-world charitable giving. That in and of itself makes this paper a big contribution. Plus, peer-to-peer fundraising has the potential to revolutionize charitable giving, so it's awesome that you're looking into it.
- On anonymous donations, I might cite Hoffman, Hilbe, and Nowak's work on "signal burying": <https://www.nature.com/articles/s41562-018-0354-z>

Reviewer #3:

Remarks to the Author:

This manuscript reports the results of a series of analyses performed upon a "big data" set (donations to the GoFundMe platform). The analyses were designed to test multiple questions in the research area of altruism and charitable giving. The central goal of this paper is to replicate previous findings on multiple altruism-related research questions. To this effect, the authors found that 21% of the donations were made anonymously—opposing the claim that altruism is exclusively altruistic.

The authors should be commended for the attempt to bring real world data to this area of research, using a large sample of real monetary donations helps move this research area beyond dictator games and self-report measures. It was also admirable that the authors were very upfront about the exploratory nature of their analyses (i.e., how several research questions and hypotheses were formulated only after receiving the data). Two additional strengths included the use of a variety of sophisticated methods such as language analysis and name-based gender predictions as well as the

high power of the statistical analyses. The paper was also well written and likely to be of interest to a broad selection of social scientists. Despite these strengths, there are some major limitations with this manuscript that would need to be addressed before it would be suitable for a high impact journal.

The central question here was whether the fact that 21% of people donated anonymously rules out egoistic motivation (H1 is that "a substantial percentage of donations will be made anonymously", but they never specify what percentage qualifies as substantial. Would 1% be sufficient? 5%? Why?). The main limitation is that it's unclear how we know whether anonymous donations are not egotistical? As I was reading this paper, I watched an episode of the Netflix show *Ozark* in which an anonymous donation was used to gain leverage over the recipient at a later time. I also know people who have made what are technically anonymous donations, only to drop this fact into conversation at cocktail parties (presumably, making the donation anonymously allows them to claim more moral credit when subsequently describing the act). Without ruling some of these explanations out, it's hard to know what to make of these anonymous donations.

Anonymous giving could also be explained by alternative egotistical accounts, such as the "warm glow". During the discussion, the authors attempt to address this alternative explanation by saying, "A motivation of achieving a warm glow (Andreoni, 1990) could be at play to some extent, but this does not seem to fully account for the extent of anonymous donations". It is unclear to the reader how this motivation could not fully account for the finding. An additional egotistical explanation would be that people might have donated to save taxes but chose to do that anonymously in order to conceal their financial status from others (e.g., to avoid envy) or to conceal their private support of political controversial causes (e.g., donating to a gay wedding). While I share the authors intuition that there are non-egoistic motivations that can drive altruism, the present conclusion is unwarranted given the current data.

More needs to be done around this issue before this paper could back up the central theoretical contribution. One possible strategy is that the authors could see if anonymous donors are more likely to "other orientated" language (i.e., more empathic language). Another is to survey people who have used the website and find out what proportion of "anonymous" donors do not ever report their donation and do not admit some ulterior, egotistical motive. If a substantial portion of the 21% either report the donations to others, make them conspicuous in some way, or admit to other egotistical motives, then this would render the current interpretation problematic.

Another limitation was the operationalization of variables and the conclusions the authors can draw from their analyses. To begin, whether someone was "kin" was operationalized as whether the donor and recipient shared their surname. I agree this is a clever, albeit it rough, operationalization. However, the authors suggest that is people who share last names, but are not related, donate this would only attenuate their relation. But there is an alternative possibility: people tend to donate more to people who are not related to them but share their last name, consciously or not. If such a relation exists, it would be impossible to distinguish it from the true kin effect of interest, and it could artificially inflate their finding.

As for the conclusions the authors draw from their results, one general issue is that they seem to be making causal claims based on correlation. For example, they suggest that larger previous donations cause individuals (especially men) to make larger donations. In reality, there are numerous plausible confounded variables that may explain this relationship (e.g., whether the cause/issue is timely, which geographic area the charity involves- and its average SES- etc.).

Another questionable interpretation is the interpretation of whether women or men donate more overall. The results were inconsistent—there were more women than men that donated, but the

average donation per individual was greater for men than for women. They come up with multiple alternative explanations/ limitations of their data, which I appreciated, but they still conclude that "however, our results do line up with the majority of past findings regarding a higher baseline generosity in female donors." It is unclear to me why this was the overall conclusion. Past laboratory studies comparing women and men on charitable donation have focused on differences between average amount donated, so I am not sure why the conclusion instead didn't, if anything, lean towards men donating more than women.

Another questionable interpretation is that of H4. The interpretation of H4 is based on a very one-sided look at the data. The data is perfectly consistent with the interpretation that females give more when relatively more males are present while males give more when relatively more females are present. This way, females and males might both show contributions as a costly signal but possibly to a varying extent. This interpretation of the data is underappreciated in the stated results section (only a very qualifying statement at the end of the section indicates that possibility). I appreciate that the costly signaling effect for women disappeared when outliers $\pm 1SD$ were excluded. This strict exclusion criteria, however, seems not appropriate to apply (in fact, I have never read a paper before in which outliers $\pm 1SD$ were removed from the analysis).

Minor issues:

- The theoretical rationale was not discussed until much later in the paper (instead, the introduction focused heavily on the dataset). It is more common to first articulate the key research questions, then explain why the current data set was selected to answer these questions. This is, of course, a matter of preference. But it would make the work more accessible to a much broader audience.
- It would be ideal to include a sensitivity power analysis given that some effect sizes are very small (e.g., a small monetary difference of contributions between women and men) or don't seem meaningful (e.g., H4).
- The language surrounding the variable of "visibly present females" was confusing. It took me until much later in the paper before I realized that the authors operationalized this by female "past donations" listed on the page and not by somehow being able to tell who was online at the same time as the individual donor. Please make sure this definition is clear as early as possible.
- H6 is stated as a mediation model and in a way that cannot be tested by the given data. This is misleading and an exaggeration of what the present data can tell us.
- It is misleading how results are presented regarding H3. The effects are not even marginally significant but are interpreted as real effects as first. The authors state only in the end that the results are not significant.
- The authors wrote: "If humans have a predisposition to help family members, we should see significantly higher contributions to recipients who are apparently related to donors than to those who are not. Our second hypothesis, H2, is that a significantly higher amount on average will be donated to recipients who are apparent family members of donors than to those who are not." I found the wording of this passage to be confusing and am unsure how the two predictions differ from each other.
- "Our further analyses of the more than 300,000 self-identified donations test..." I am not sure what "self-identified" means in this context.

Notes to all reviewers: we have specified below the line numbers where you can find all changes implemented in the revised manuscript. We also attached a PDF of the manuscript with track changes visible for a complete account of changes made to the original version. In the updated Supplemental Materials A PDF we presently note for your convenience any new additions in the table of contents.

Reviewers' comments:	Our responses:
Reviewer #1 (Remarks to the Author):	
Review of Sisco and Weber Charitable Giving This manuscript reports on a big data study of online charitable donations made via the GoFundMe platform. The total sample of donations was more than half a million and the total amount donated over \$40 million. For some comparisons only a rather smaller subsample of the data was usable, but the sample size was still quite large relative to purposed collected experimental samples. The study is limited by the nature of the online data retained by GoFundMe. For example, demographic data on donors and recipients is limited to first and last names (unless donations were anonymous). The authors utilized first names to characterize gender and last names to check for kin based giving. Their procedures for using names in lieu of more direct demographic data are sensible and they are candid about their resulting imperfections. They make a strong case for observational studies, experiments, and questionnaires as having complementary methodological strengths and weaknesses. When the findings of multiple methods are consistent we can be rather more confident of them than if we are reliant on only one.	We thank Reviewer #1 for her/his thoughtful comments and suggestions on our manuscript. We have closely reviewed the literature suggested and revised the manuscript accordingly. Below you can find our detailed responses to each comment by Reviewer #1 with accounts of what actions we took to address them.
The headline result of this study is that 21% of GoFundMe are made anonymously. This is significant because there is a large controversy in the literature over whether all cooperative behavior can be cashed out in the end in terms of individual costs and benefits or whether there is a significant amount of genuine selfless altruism in human behavior. The authors take their results to be robust support for the later alternative. I think they are correct.	We note that in response to the other two reviewers we have collected more data which strengthens the robustness of this claim. In short, we conducted a survey of past GoFundMe donors. We asked donors a battery of questions regarding their motivations for making past donations. The results conservatively suggest that at least 11% of anonymous donations are genuinely altruistic. Thus we now offer this smaller percentage of donations in our dataset as robust evidence of truly altruistic donations.
The authors also find support for kin altruism being important, for a sexual selection hypothesis, and for a social influence effect. They do not find support for the idea that women give more than men but women do	

express emotional motivations for giving more than men. These findings are all consistent with previous reports from experimental and questionnaire based studies and with theoretical expectations. Thus this study does find consistent support for several important hypotheses.	
The theoretical discussion (lines 155-294) leading to the 6 hypotheses the authors' test is not as strong as it should be. The authors don't seem to be aware that the evolutionary theory of altruism has progressed far beyond Trivers and Hamilton, as important as these classical insights are. Steward-Williams (2015) is a decent review.	We thank Reviewer #1 for suggesting this review paper and pointing out that we could strengthen our theoretical discussion of altruism. We have revised the introduction (lines 163-307) to more comprehensively discuss evolutionary theories of altruism including notable developments since Trivers and Hamilton. In line with more emphasis on evolutionary theories we have revised the manuscript throughout to discuss theoretical work as a third category of relevant past work in addition to lab and field studies.
The authors might want to briefly situate their findings in the larger literature on variation in prosocial behavior. The very large experimental literature on economic games is interesting in that in such games as the Public Goods Game with punishment, cooperation in many samples is nearly perfect after ten rounds if participants are given tools like punishment or cheap talk to encourage cooperation (Baum et al. 2012). In unpublished work in my lab it seems like on the order of 20-30% of participants act as altruistic leaders in establishing cooperation. They cooperate themselves, use whatever tools like punishment and persuasion they are given to encourage a reluctant majority to cooperate. I think other peoples' experience is similar. It is also worth noting that prosociality is highly variable cross-culturally (Henrich et al. 2010).	We agree with this suggestion and have added the suggested paper (Baum et al. 2012) in the conclusion section (lines 858-859). We feel the finding that public goods games have much more cooperation when communication is involved supports a statement in our conclusion regarding why it is important whether or not humans sometimes act without self-benefiting goals. We also find connecting our paper to the literature on cross-cultural variation (e.g. Herrmann et al. 2008; Henrich et al. 2004), to be a helpful suggestion. We found also with this literature that making a connection in the conclusion was appropriate. We now point to more cross-cultural investigations of a similar type as the current paper as a promising future direction (lines 866-868).
For example, Herrmann et al. (2008) using the PGG with punishment discovered a number of societies in which anti-social punishment was common. In these groups cooperation rates remained low. Both the Henrich and Herrmann used observational data to argue that their findings had external validity. The substantial minority of altruists Sisco and Weber found seem to be important out of proportion to their numbers.	We thank Reviewer #1 for bringing up this potential inconsistency between our results and those of Henrich et al. (2004) and Herrmann et al. (2008). We have read these papers and can offer a few reasons why we feel that there is not a notable inconsistency upon closer inspection of how their results align with ours. Firstly and most importantly, we would like to clarify that the anonymous donations we find in our data do not necessarily contain the only altruistic donations. We do not mean to argue that the rest of donations are selfishly motivated. Rather, we suggest that our anonymous donations are the only ones for which we can most confidently rule out selfish motivations. The percentage that we suggest are only attributable to altruistic motivations are a conservative lower bound estimate of the percentage of truly altruistic donations in our dataset. We feel it is likely that some non-anonymous donations are truly altruistic, but we cannot as strongly rule

	out egoistic motivations for them. We explain this now in the discussion in lines 718-723. Secondly, we believe that the games in the Henrich et al. and Herrmann et al. studies are not quite analogous to the decisions in our data. Henrich et al. (2004) look at a Public Goods Game in which decisions can be driven by altruism and/or strategic cooperation. As with most PGGs, we cannot disentangle altruistic motivations from strategic self-oriented motivations behind participants' decisions in the PGGs of Henrich et al. (2004). In our data, the decisions arguably do not involve a strategic cooperative component such as in a PGG as it is not a public good that donors are contributing to with most GoFundMe campaigns. In the Herrmann et al. (2008) paper they use an ultimatum game. The authors focus on the behavior of the recipients and whether or not they choose to engage in altruistic punishment. This does arguably provide a measure of altruistic behavior isolated from strategic motivations given that they were one-shot games. However, we feel this is still not an analogous type of altruistic decision to the ones in our data. Punishing an unfair allocator in an ultimatum game can be construed as altruistic in that it aims to benefit society, but this is different in several ways from helping one person or a specific group of people such as in the case of our donations. In sum, we offer that a.) our results do not rule out that a much higher percentage of altruists might exist in our data, and b.) decisions in PGGs and ultimatum games are not directly comparable to decisions in our dataset.
Stewart-Williams, Steve 2015 Morality: Evolution of. In International Encyclopedia of the Social & Behavioral Sciences edited by J. D. Wright, pp. 811-818. 2nd ed. vol. 15. Elsevier, Oxford. Baum, William M., Brian Paciotti, Peter Richerson, Mark Lubell and Richard McElreath 2012 Cooperation due to cultural norms, not individual reputation. Behavioural Processes 91(1):90-93. Herrmann, Benedikt, Christian Thöni and Simon Gächter 2008 Antisocial punishment across societies. Science 319:1362-1367. Henrich, Joseph, Robert Boyd, Samuel Bowles, Colin Camerer, Ernst Fehr and Herbert Gintis 2004 Foundations of Human Sociality: Economic Experiments and Ethnographic Evidence from Fifteen Small-Scale Societies. xix + 451 ed. Oxford University Press, Oxford.	All of these papers have been reviewed and included in the manuscript.

Reviewer #2 (Remarks to the Author):	
Nature Comms Report I apologize about the delay in providing this report. I had several other reports that I needed to complete first and since I am also teaching this semester, it took me a while to do so. I know such a delay can be frustrating for the authors. Thanks for your understanding. The authors analyze a large dataset of GoFundMe donations. They find that some 20% of donations are anonymous, that donors give more to family members, that donors give 5c more for every additional dollars others have given. They also find interesting gender differences: men respond more to the amount others have given, and give more when a higher proportion of other donors are female. People’s justification of why they give also varies by gender.	We thank Reviewer #2 for her/his thoughtful comments on our manuscript and the suggestions for alternative framings. We have carefully considered each of Reviewer #2's comments and revised the paper accordingly. Our detailed responses to each comment by Reviewer #2 and accounts of specific actions we took to address them can be found below.
This paper makes a substantial contribution by analyzing such an extensive dataset of real-world contributions. The findings meaningfully help to advance our understanding of altruism. Indeed, I believe that with some revision, this can be a first-order contribution. However: (1) I strongly disagree with the current framing. I will try to explain why and will suggest an alternate framing. (2) The paper would benefit from extending the analysis of donations to kin vs. non-kin, and focusing more on this.	To briefly summarize our revisions regarding Reviewer #2’s two overarching comments, (1) we have revised the framing substantially to clarify how our hypotheses are related to proximate vs. ultimate drivers of altruism. (2) We conducted all of the kin vs. non-kin analyses suggested by Reviewer #2 that were possible with our data and include these in the manuscript where the results seemed appropriate and robust. We have placed more emphasis on the kin vs. non-kin analyses in the paper where we could; but feel it would not be best to primarily focus on kin vs. non-kin because of the limitation that some uncertainty is involved in using same last names as a proxy for kin relations. We still discuss the hypothesis and claim regarding true altruism, however it has been make more convincing in at least two ways. Firstly, we have substantially clarified the question with the discussion of proximate vs ultimate motivations. Secondly, we designed and implemented a survey of past GoFundMe donors to ascertain their motivations behind giving. The results of this survey strengthen the claim that a significant percentage of our donations were motivated by genuine altruism (at the proximate level).
First, my main criticisms regarding the framing:	
- I would cite the literatures on norm enforcement and indirect reciprocity. I like to cite Boyd (Ch. 2 of “A Different Kind of Animal” for instance) on norm enforcement. I typically cite Nowak’s “five mechanisms” on indirect reciprocity. But there are lots of options.	We appreciate this suggestion to cite more work on evolutionary theories of altruism (which was also echoed by Reviewer #1). We have included a discussion of cultural group selection (Boyd & Richerson 2009) based on this recommendation in lines 299-307. We also now discuss indirect reciprocity (lines 279-283) as part of a more

	comprehensive review of evolutionary theories of altruism. We found the suggested citation of Nowak 2006 to be an appropriate one to include alongside a few others that we have added to the manuscript.
- The argument confuses “proximate” (what we think and feel) and “ultimate” explanations (what shaped those ideologies and preferences). For instance, consider this sentence, “Do humans take costly actions to help others for exclusively egoistic reasons, or are other-regarding goals also at play?” That’s a false dichotomy. We typically think of “other-regarding goals” as likely having developed for reasons that, ultimately (though not necessarily consciously) benefit the individual. That is the contention inherent in the norm enforcement and indirect reciprocity literatures. Of course, that doesn’t mean we consciously deliberate over the egoistic benefits of donating. Partially this is because, in general, we often aren’t aware of the motivations that drive our ideologies and preferences. We historically had no idea that cheesecake tasted great because it was a calorie bomb—we just found sweet and fatty foods tasty. We don’t typically deliberate over the fertility of a potential mate, we just find them hot. Etc. And, partially, it’s because in the domain of altruism, people benefit from being seen as someone who doesn’t deliberate over the benefits to themselves. For more details see http://www.pnas.org/content/112/6/1727. For instance: “First, psychologists and philosophers have long asked the following question: is helping others “always and exclusively motivated by the prospect of some benefit for ourselves, however subtle” (20) [for example, the conscious anticipation of feeling good (21), avoidance of guilt (22–24), reputational benefits, or reciprocity (1–14)]. At the extreme, this question amounts to asking if saintly individuals, such as Gandhi or Mother Teresa, were motivated thus or if they were authentic altruists who did good without anticipating any reward and would be altruistic, even in the absence of such rewards. Our model suggests that authentic altruism is, indeed, possible: by focusing entirely on the benefits to others, authentic altruists are trusted more, and the benefits from this trust outweigh the risk of, for example, dying a martyr’s death.” (To be clear, I don’t think you need to cite this paper. I just hope it helps to clarify the issues raised by your existing framing.)	Reviewer #2's point about our original manuscript conflating "proximate" with "ultimate" explanations of altruism is well-taken. We understand her/his point and acknowledge that from an evolutionary perspective, even if a person is consciously thinking they are doing something with the benefit of the recipient as her ultimate goal, the ultimate reason she has a tendency for this is likely because the trait was selected. Somehow this tendency increased the likelihood of ancestors' genes propagating forward, and therefore ultimately altruistic behavior exists (evolved) because benefits accrued to the individual. We feel, however, that there is still a relevant research question about the psychological (proximate) motivations experienced by the individual. In the interesting paper that Reviewer #2 mentioned, (as Reviewer #2 quoted) it says: “First, psychologists and philosophers have long asked the following question: is helping others “always and exclusively motivated by the prospect of some benefit for ourselves, however subtle”.” This is indeed the question that we feel is relevant to address in part with the current dataset. The question of if, as experienced by the individual, there are ever cases where individuals are motivated with the ultimate goal of helping another person rather than the ultimate goal of bettering themselves has been pondered for centuries. We feel our data provide meaningful evidence and help answer this question about the psychological (proximate) reasons for giving. The second quote that Reviewer #2 offers states that the extreme version of this question is if persons like Gandhi were ultimately motivated by selfish goals or not. This point is well-received by us. Personally, we agree that it seems counter-intuitive to suggest that such benevolent persons could have been entirely motivated by selfish goals. However, we believe that proponents of the view that purely altruistic motivation is impossible in humans would argue that in fact Gandhi may have been ultimately seeking self-benefiting goals such as the respect he gained by helping so many people, the personal pride it delivered him, or selfish goals about his name becoming a symbol of virtue. We agree with Reviewer #2 that it seems humans are capable of behaviors motivated primarily by helping others. But, we offer that the debate over whether or not this capacity exists (at the psychological/proximate level) has been ongoing for quite some time and is still relevant. We feel our data make a

	valuable contribution to this long-debated question. In order to address this comment by Reviewer #2 we have substantially revised the manuscript to clearly differentiate evolutionary vs. psychological levels of altruism as can be seen in lines 170-175 and more broadly in our discussion of psychological and evolutionary perspectives in lines 163-307. We also revised the language related to hypothesis 1 throughout the paper to clarify that it pertains to proximate, not ultimate, reasons for altruism.
Here is how I would, instead, frame your contribution. (Notice the more central role of the kin vs. non-kin contrast.)	We performed all of the analyses suggested regarding kin vs non-kin, and have included them in the manuscript where we felt they were appropriate and robust. As stated above, we were happy to expand the emphasis on kin-related analysis, however we feel that what we have done is as much as we can emphasize kin vs non-kin analyses in the paper given that the stated limitations of family name matching and the relatively small sample size of donations where donors and recipients share the same last name.
- A large literature identifies reputations as a major driver of altruism towards non-kin. Cite the literatures on norm enforcement and indirect reciprocity. Maybe also cite the costly signaling paper.	We now discuss reputations in the introduction (lines 279-283) and discussion (lines 695-698) sections. We discuss costly signaling primarily in lines 367-383 and 790-805.
- Emphasize that this is not necessarily conscious.	We now emphasize that costly signaling is not necessarily conscious in lines 797-805.
- This is thought to explain many otherwise puzzling features of altruism, like why people care so much if they are being observed (cite the observability lit), why they don't attend so much to effectiveness (cite the scope insensitivity literature), and why they attend so closely to normative information.	On citing observability literature. We now cite a meta-analysis of studies on prosocial behavior involving observability in the discussion section (lines 733-738). We discuss the finding that anonymous donors gave less than public donors which lines up with the results of the meta-analysis. On citing scope insensitivity literature. As Reviewer #2 suggested below, we conducted an analyses focused on how amounts change depending on closeness to a campaign's goal. As explain below we found these analyses unfortunately to be inconclusive due to uncertainty about the relevant goal amounts perceived by donors. As analyses related to scope insensitivity do not seem appropriate for inclusion in the paper, we did not include a discussion of scope insensitivity in the manuscript. On attending to normative information. We believe we understood this suggestion, but would be happy for Reviewer #2 to clarify it if we did not. To our minds, discussing reasons that participants may be attending to normative information would primarily include reviewing judgment and decision making theories like anchoring and adjustment, query theory, prospect theory, and default effects as well as social psychological accounts such as the influence of social norms.

	While we feel it could be quite interesting to speculate about why donors in our sample seem to be influenced by normative information (i.e. amounts donated by previous donors), we do not feel that our data provide information on the underlying mechanisms behind the behaviors. In other words, we could discuss theories of how normative information influences people, but we cannot argue that our data show evidence of one process more than another. In light of this, we feel that there is just not room for a discussion of this topic given that the manuscript is already longer than the suggested word count for Nature Communications articles.
- This paper is the first to provide an analysis of a substantial number / \$\$ value of real-world donations and show that the patterns of giving are consistent with these predictions of these literatures.	We do make a point of this.
- A unique feature of this dataset that makes it especially useful for this purpose is the the ability to observe and contrast the donation behavior of kin and non-kin. Critically, since altruism to kin is driven not by reputations, but by kin selection, the puzzling features discussed above are expected to be weaker for kin.	As we explain more below we conducted the kin interaction analyses and did incorporate them into the paper.
- The key findings are:	
- The vast majority of people don't give anonymously. When they do give anonymously, they give less	We have added a more theoretical discussion of this now on lines 733-738.
- People give substantially to kin, both in terms of number of donations and their avg size.	We do make a point of this.
- Ideally, here you would discuss that people aren't so sensitive to recipient's need (see below); but are much more sensitive when giving to kin.	We appreciate Reviewer #2's suggestion for this additional analysis. It is a very interesting idea, but we find it to be infeasible with our data. The issue we found with trying to do this analysis is that we have a prohibitively small sample size of donations made near the campaign total amounts with donors sharing the same last names as recipients. We already have noted in the manuscript that filtering the dataset down to only donors with the same family names as recipients reduces the size of the data down substantially. Cutting it down even more by only looking at "same name" donations made near the campaign goal amount results in an underpowered sample size of 198 donations. Therefore, due to limitations of the data unfortunately we cannot implement an analysis of if people are more sensitive to the needs of kin as Reviewer #2 suggested.
- People respond substantially to normative information about what others give. But, less so when giving to kin.	We thank Reviewer #2 for this interesting suggestion. We completed this analysis by running a regression with an interaction between Same_last_name and Mean_visible_donation. The results of this analysis can be seen in our updated Supplemental Materials A in section 5.5. Interestingly the result was the opposite of what Reviewer #2 proposed. We see a positive and significant interaction with these two variables such that it appears when giving to kin

	our donors were more sensitive to normative information. We understand Reviewer #2's reasons for expecting that people would be less sensitive when giving to kin, that they are more motivated to actually help kin so comparisons to others' donations should matter less. However, after reflecting on the results of this analysis, it also seems reasonable that people could react more to normative information when giving to kin. If one was donating to his brother for a medical treatment, we would expect that he is truly motivated to help his brother. Additionally, however, he may be aware of the impact this donation has on his reputation and relationship with his brother. The social norm is that people should be more generous toward kin than non-kin. In light of this, others and his brother may be expecting a higher-than-average donation and therefore if a donor gave less than the normative amount his reputation may be more damaged than if the lesser donation was to non-kin. Since these are very exploratory findings hinging on a small subset of our sample, we discuss the exploratory analysis now in the discussion section (lines 750-758) and we mention that readers can find more details and the regression table in the Supplemental Materials A5.5.
- There are even some gender differences that conform to this hypothesis:	
- Men give more - Men are more "competitive" in their giving, giving more when more others give - Men give more when more females observe the gifts	Yes, these are part of the manuscript.
- Ideally, you'd show these effects go away for kin	Interestingly, as we described above the effect of normative information appears to get stronger when giving to kin (See SM A5.5 for the regression table). The effect of visible females is also significantly stronger when giving to kin (also shown in SM A5.5). We feel that essentially the same explanation as given above applies: strong social norms that one should be more generous to kin would make it especially important to donate generously to kin when the opposite sex is more present.
- In addition, there are some interesting gender differences in how people justify donations	
I suggest the following additional analyses that would support this framing, and, IMO, generally improve the paper:	

- Can you please run an interaction between `same_last_name` and `mean_visible_donation`? You should find that the effect of `mean_visible_donation` is smaller for kin.
- Similarly, `prop_visible_female` should matter less for kin. (Basically, I would interact kin with everything, and showing that it always reduces your coefficient.)

As described above, we completed these analyses and the results can be seen in SM A5.5. Both of the interaction effects were positive and significant.

- Can you show that people are generally not-so-responsive to need somehow? I.e. show that they give when lots of others are giving, even if the campaign doesn't need it?

We also appreciate this interesting idea for an additional analysis. We did conduct the analysis suggested here. The visualized trajectories of average donation amounts as the goal is approached can be seen below.

This graph shows the average amounts donated on the y axis and the percent of the goal met *before each donation* on the x axis. Thus, any dots after 1.0 on the x axis were made after the goal amount was reached.

While this is surely an interesting idea for an analysis with our data, we find it is actually difficult to meaningfully interpret. We see that male, female, and anonymous donors all start giving less after the goal is met. That makes sense. But, are these drop-offs slow enough to be thought of as “not-so-responsive to need”? What slope would be an appropriate cutoff?

We feel any cutoff threshold we could set could only be arbitrary and subjective. More importantly, we feel that we cannot assume the goal amount on the campaign page is what donors perceived the goal of the campaign to be. Imagine that a campaign page described a surgery that costs \$100k; say they asked for \$10k as a goal but explained that every dollar can help. Donors would likely perceive \$100k as the true goal and therefore it would be inaccurate to analyze the data assuming \$10k was the true “goal amount”.

Since we find that we cannot interpret these results robustly we feel they are not fit to be included in the manuscript.

And... can you then show that kin tend to be more sensitive to the recipient's need than non-kin?	As explained above, we do not have a sufficient sample size of kin donations near campaign goal amounts to conduct this analysis. We can still generate the visualization (below) but we feel it is not appropriate to try to draw conclusions from it given the small sample size. - Can you tell what proportion of a given recipient's gifts came from kin?	No, we cannot confidently estimate this. Assuming that when donors and recipients share the same last name they are related is pretty safe, but we do not know how many donations were made by kin that do not share the same last name with recipients.
- Ideally, you'd be able to say more about what's different about anonymous donations. Do you have the data to analyze who the donors were? Or is all you can see is a line of data with an anonymous donation?	Given the limitations of our data we cannot say any more about the anonymous donations. For anonymous donors we only know what campaign they donated to, how much they donated, and if they left a message (very few did).
	
Here are some additional suggestions and comments:	
	
- I had a bit of trouble telling what your data look like. Perhaps you can clarify this further? (In particular, as you can tell from the above, I would like to see some analyses at the recipient level, but I can't tell if you can do this using your data and are constrained to the particular variables in Tbl. 1.	We have uploaded our dataset with the current revision. All of our analyses can be replicated with that dataset and the R code provided in the Supplemental Materials A document. We have not included the names columns for confidentiality of the donors and recipients. To the same effect, before we make this dataset public, we will scrub all names from the messages. We have not done this in the current dataset and therefore we ask that reviewers do not share it with anyone in its current form.
Another question: Is the only information donors can see the "visible" donations, or can they dig around and get more info? Is that the only info you can see?)	Donors were shown the past 10 donations on the page at the time of their donation decisions, but they could browse more past donations if they wanted to.

- A glaring omitted cite is to Croson and Gneezy’s review of gender differences in charitability. I advise familiarizing yourself with this paper and the literature in it, and framing your contribution on gender differences accordingly. This comment is intended to be constructive, as I think you add a lot by showing that the magnitude of gender differences is large in this important, real-world setting.	We thank Reviewer #2 for suggesting this paper. After looking closely at this review we did find two additional papers (Houser & Schunk and Bolton & Katok) that used dictator games to report on gender differences which we have now added to our manuscript along with mentioning the Croson and Gneezy review itself. The other gender differences discussed in the review (inequality aversion, trust, reciprocal behavior, cooperation in social dilemmas) we feel are not quite related enough to be discussed in our manuscript because there is not a way we can test them with our data. If Reviewer #2 has ideas for how some of these other gender differences could be analyzed with our data we would be happy to hear her/his thoughts, but we do not see any possibilities with those ourselves.
- Tbl. 1 is woefully inconsistent with the norms of general science journals and should be reformatted. E.g., use English variable names, and add a figure legend that allows the table to be interpreted on its own. What is the interpretation of the coefficients? What units are the variables in? Etc.	We have revised Table 1 to include English variable names and a descriptive figure legend addressing the questions Reviewer #2 mentioned.
- IMO, you’re not making enough of the setting. There are so few analyses of real-world charitable giving. That in and of itself makes this paper a big contribution. Plus, peer-to-peer fundraising has the potential to revolutionize charitable giving, so it’s awesome that you’re looking into it.	We thank Reviewer #2 for this suggestion. We feel the last paragraph of the discussion makes this point (lines 862-868).
- On anonymous donations, I might cite Hoffman, Hilbe, and Nowak’s work on “signal burying”:https://www.nature.com/articles/s41562-018-0354-z	We found this suggested paper to be quite useful in evaluating the potential motivations of the anonymous donors. We now discuss this paper and the possibility of donating anonymously as an instance of signal burying throughout the manuscript.
	
Reviewer #3 (Remarks to the Author):	
This manuscript reports the results of a series of analyses performed upon a “big data” set (donations to the GoFundMe platform). The analyses were designed to test multiple questions in the research area of altruism and charitable giving. The central goal of this paper is to replicate previous findings on multiple altruism-related research questions. To this effect, the authors found that 21% of the donations were made anonymously—opposing the claim that altruism is exclusively altruistic.	

The authors should be commended for the attempt to bring real world data to this area of research, using a large sample of real monetary donations helps move this research area beyond dictator games and self-report measures. It was also admirable that the authors were very upfront about the exploratory nature of their analyses (i.e., how several research questions and hypotheses were formulated only after receiving the data). Two additional strengths included the use of a variety of sophisticated methods such as language analysis and name-based gender predictions as well as the high power of the statistical analyses. The paper was also well written and likely to be of interest to a broad selection of social scientists. Despite these strengths, there are some major limitations with this manuscript that would need to be addressed before it would be suitable for a high impact journal.	We thank Reviewer #3 for her/his supportive words about the manuscript. We have carefully considered on all of the points that Reviewer #3 suggested we address and describe our specific responses to each comment below.
The central question here was whether the fact that 21% of people donated anonymously rules out egoistic motivation (H1 is that “a substantial percentage of donations will be made anonymously”, but they never specify what percentage qualifies as substantial. Would 1% be sufficient? 5%? Why?). The main limitation is that it’s unclear how we know whether anonymous donations are not egotistical? As I was reading this paper, I watched an episode of the Netflix show Ozark in which an anonymous donation was used to gain leverage over the recipient at a later time. I also know people who have made what are technically anonymous donations, only to drop this fact into conversation at cocktail parties (presumably, making the donation anonymously allows them to claim more moral credit when subsequently describing the act). Without ruling some of these explanations out, it’s hard to know what to make of these anonymous donations.	We thank Reviewer #3 for this point, it is well-received. Based on a recommendation farther below by Reviewer #3, we designed and conducted a survey of people who previously made donations on GoFundMe. We asked those whom had made anonymous donations a battery of questions (shown in Supplemental Materials B) designed to rule out all possible egoistic goals that might have been driving their donations. We included all of the possibilities that Reviewer #3 brought up and all others we could find in the past literature. We found that 11% of our respondents whom had made anonymous donations responded "no" to each of the potential egoistic motives that were asked of them. 53% of all GoFundMe donors we surveyed reported their ultimate motivation was helping the recipient. We have now incorporated these results into the paper. We clarify that by a "substantial" percentage we mean one that is statistically greater than zero after accounting for the percentage of anonymous donations we estimate were made for selfish reasons. This results in an estimate that 11% (SE=2.4%; $p < .001$) of anonymous donations were made without any self-benefiting goals attributable to their motivation. As 11% (+/- 4.8%, 95%CI) is statistically significantly different from zero, we find robust support for our first hypothesis after incorporating supplementary survey results. The survey procedure and results are now reported primarily in lines 453-470 and lines 484-516.

Anonymous giving could also be explained by alternative egotistical accounts, such as the “warm glow”. During the discussion, the authors attempt to address this alternative explanation by saying, “A motivation of achieving a warm glow (Andreoni, 1990) could be at play to some extent, but this does not seem to fully account for the extent of anonymous donations”. It is unclear to the reader how this motivation could not fully account for the finding. An additional egotistic explanation would be that people might have donated to save taxes but chose to do that anonymously in order to conceal their financial status from others (e.g., to avoid envy) or to conceal their private support of political controversial causes (e.g., donating to a gay wedding). While I share the authors intuition that there are non-egotistic motivations that can drive altruism, the present conclusion is unwarranted given the current data. More needs to be done around this issue before this paper could back up the central theoretical contribution.	We thank Reviewer #3 for these suggestions of egoistic self-rewards that might drive participants to make anonymous donations. To address them, we included giving “to feel better about oneself”, “...to feel good”, and “...to feel pride” in our battery of questions about why users made anonymous donations. We also asked about donating to save on taxes as Reviewer #3 suggested. The full battery of questions can be seen in Supplemental Materials B. Again 11% of our anonymous donors reported that none of these or any other egoistic goals were at play when making their donations. It’s helpful to distinguish between why people donate, and why they choose to do it anonymously. The reasons we discussed above on this page regard why people donate. This is the essential question we are focused on with hypothesis 1. Reviewer #3 also mentions some reasons why one would choose to donate anonymously. These include to conceal financial status or to conceal support of controversial causes. This is a separate question that we feel is separate from the core motivation behind donating. (The decision to make a donation anonymously or publicly does not affect the ultimate motivation behind the decision and therefore we feel is not relevant enough to our key question to be included in the manuscript. If someone’s ultimate goal is to help someone in need, and they choose to do it anonymously to conceal their financial status, still their motivation for helping was altruistic. If a person’s ultimate goal is to receive a tax reduction, and they choose to do it anonymously to conceal their financial status, still their motivation was ultimately about a self-benefit. In other words, the decision to make a donation anonymous or not is interesting, but we believe is distinct from the ultimate motivation behind the donation (truly altruistic or self-benefiting).)
One possible strategy is that the authors could see if anonymous donors are more likely to “other orientated” language (i.e., more empathic language).	This is an interesting idea but very few anonymous donors (<1%) left messages, so this analysis is just not feasible with our data.
Another is to survey people who have used the website and find out what proportion of “anonymous” donors do not ever report their donation and do not admit some ulterior, egotistical motive. If a substantial portion of the 21% either report the donations to others, make them conspicuous in some way, or admit to other egotistical motives, then this would render the current interpretation problematic.	We thank Reviewer #3 for this very helpful suggestion. We implemented a survey based on this recommendation which we described in detail already above.

Another limitation was the operationalization of variables and the conclusions the authors can draw from their analyses. To begin, whether someone was “kin” was operationalized as whether the donor and recipient shared their surname. I agree this is a clever, albeit it rough, operationalization. However, the authors suggest that is people who share last names, but are not related, donate this would only attenuate their relation. But there is an alternative possibility: people tend to donate more to people who are not related to them but share their last name, consciously or not. If such a relation exists, it would be impossible to distinguish it from the true kin effect of interest, and it could artificially inflate their finding.	We feel that Reviewer #3 made a valid point here and we thank her/him for this suggestion. How much might donors who by chance share the same last name as recipients affect our results? In order to investigate this, we performed a simulation to quantify the likelihood of donors and recipients sharing a last name by chance. This robustness check is now mentioned in a footnote on line 261 of the paper and can be found in Supplemental Materials A7. We find that when we randomly reshuffle which donors are matched with which recipients there are only about 110 matches on average, while in our data we see 5,144 matches. This suggests that chance alone would produce about 2% of the same name matches we see in our data. It seems intuitive that having about 2% of the same name matches in our data due to chance alone would not affect the final results very much. To be sure, we evaluated this quantitatively to ascertain the potential for bias due to 2% chance matches. We find (seen in SM A7) that for chance same name matches to entirely create the effect of \$29.27 we see in our main results, chance same name donations would need to be about \$1,344 more on average. If chance same name matches gave \$29.27 more on average this would increase our estimate of the kin effect by about \$0.64. We demonstrate and note this in SM 7 and note it on page 7 of the manuscript.
As for the conclusions the authors draw from their results, one general issue is that they seem to be making causal claims based on correlation. For example, they suggest that larger previous donations cause individuals (especially men) to make larger donations. In reality, there are numerous plausible confounded variables that may explain this relationship (e.g., whether the cause/issue is timely, which geographic area the charity involves- and its average SES- etc.).	We thank Reviewer #3 for carefully considering potential confounds in our analysis. We believe that the suggested confounds are not substantial concerns for our analysis. We have prepared some statistical simulations and additional analyses to support this argument. 1.) Some of Reviewer #3’s suggested confounds are at the campaign-level. Location or popularity of a campaign could make the donations to it on average more or less compared to other campaigns. This is a sensible concern, and our analysis effectively controls for this by estimating a random effect for each campaign as well as for each campaign category. This approach is quite similar to putting in a dummy variable for each campaign which would fit a beta estimate to account for the average donation amount given to each campaign. With average amounts per campaign controlled for, other estimated effects such as the influence of recent donations are above and beyond (or “controlling for”) the campaign-level differences.

(Technical note: fitting campaigns as random effects instead of dummy variables forces the distribution of campaign estimates to follow a normal distribution and "partially pools" the information across campaigns. This means that campaigns with very few observations receive an estimate fitted with more influence by the mean of all campaign effects and campaigns with larger numbers of observations are less influenced.)

Ultimately, because we include an effect for each campaign in our regression, we control for any campaign-level effects such as if the issue is timely or if the geographic area has higher SES.

To demonstrate this, we prepared a simple simulation now in Supplemental Materials A8. We demonstrate there the effectiveness of our method of controlling for campaign-level effects that we just described above. It is a thoroughly commented simulation so as to be self-explanatory for readers familiar and unfamiliar with the programming language it was written in (R).

In this simulation, we created a dataset where we simulate donations to campaigns where previous donations do not have an influence on future ones, and we give campaigns different average donation amounts. We show that analyzing these data with a regression that does not control for campaign effects does skew the results, as Reviewer #3 expected. However this is not what we do in our analysis. When control for campaign-level effects (estimate random effects for campaigns) the bias disappears as we would expect it to.

2. While campaign-level effects are controlled for in our analyses (as we just described), Reviewer #3's comment also brought up the possibility of time as a confound within campaigns. For example, it could be that there are some days when people are more generous in general, such as holidays. This could force some close-in-time donations to correlate with each other which could make it appear that recent donations are influencing consecutive ones.

Intuitively, it seems unlikely that generosity has enough systematic and widespread variation over time for this to substantially affect our results. Nonetheless, we investigated the possibility of this. We reason that if the effects of the past 10 donations (the donations on the screen at the time of each donation decision) are actually due to time confounds (or any other confound) then the influence of donations just after the threshold of 10 should be very similar.

If the correlation with past donations is truly due to their visibility, then we would expect to see a marked difference in

the correlation between past donations up to the past ten and then a marked decline in correlation after that point. (This is essentially borrowing the logic behind a "regression discontinuity design" en.wikipedia.org/wiki/Regression_discontinuity_design).

You can see below that our findings are what we expected. There is an obvious difference between whether recent donations were shown on the screen or not (≤ 10 compared to > 10 donations prior).

The visualization above shows the beta estimates for each prior donation from 1-20 donations prior with amount donated as the outcome variable. For robustness we repeated this analysis with ordered clusters of prior donations (1-3, 4-6, 7-9, etc) which you can see below. We find the same pattern.

These analyses support our claim that the effects we find of recent donations are due to the visibility of these donations and not attributable to a confound of time. We have now included these robustness checks in Supplemental Materials A9.

Another questionable interpretation is the interpretation of whether women or men donate more overall. The results were inconsistent—there were more women than men that donated, but the average donation per individual was greater for men than for women. They come up with multiple alternative explanations/ limitations of their data, which I appreciated, but they still conclude that “however, our results do line up with the majority of past findings regarding a higher baseline generosity in female donors.” It is unclear to me why this was the overall conclusion. Past laboratory studies comparing women and men on charitable donation have focused on differences between average amount donated, so I am not sure why the conclusion instead didn’t, if anything, lean towards men donating more than women.	We understand Reviewer #3's hesitancy to interpret our results as suggesting that females acted more generously. We made this statement based on the assumption that males and females use the GoFundMe platform about equally. After more reflection based on Reviewer #3’s comment, we agree this may be too strong of an assumption to make. We have now revised the manuscript to simply state what the descriptive results were for the average and total amounts without a conclusion that the results support women being more generous. We feel that we cannot make the conclusion that men are more generous based on to their higher average contributions for two reasons: 1.) we do not know if women each gave to more campaigns than men did, which would cast their average donation in a different light (as the sum of donations per female donor would be higher); and 2.) men's donations may be truly higher on average, but if the men in this sample have higher incomes than the women it may be unfair to make a direct comparison of their donation amounts. A comparison representing donations as a percentage of each individual's income would allow for stronger conclusions, but we do not have this demographic information about the donors in our dataset.
Another questionable interpretation is that of H4. The interpretation of H4 is based on a very one-sided look at the data. The data is perfectly consistent with the interpretation that females give more when relatively more males are present while males give more when relatively more females are present. This way, females and males might both show contributions as a costly signal but possibly to a varying extent. This interpretation of the data is underappreciated in the stated results section (only a very qualifying statement at the end of the section indicates that possibility). I appreciate that the costly signaling effect for women disappeared when outliers +- 1SD were excluded. This strict exclusion criteria, however, seems not appropriate to apply (in fact, I have never read a paper before in which outliers +- 1SD were removed from the analysis).	Again, we thank Reviewer #3 for her/his careful consideration of our results and interpretations. We do agree after considering this comment that we can offer this finding more strongly than as "a possibility". We have revised the manuscript accordingly on lines 644-647 and 794-796.
Minor issues:	

 • The theoretical rationale was not discussed until much later in the paper (instead, the introduction focused heavily on the dataset). It is more common to first articulate the key research questions, then explain why the current data set was selected to answer these questions. This is, of course, a matter of preference. But it would make the work more accessible to a much broader audience. 	We thank Reviewer #3 for this suggestion. We originally wrote the paper with the research questions detailed first along with a review of the relevant literatures, and then described the dataset. However we and our peers who provided feedback found that this did not read as well as we expected and thus we feel the current order is ideal. Since our research questions are more diverse than usual studies, as they were generated based on what could be answered with the data and not conceived of beforehand (as we detail in the paper), we find that overviews of the dataset before introducing the research questions provides a helpful context for readers to more immediately see the relevance of our questions and their associated literatures. To clarify this for readers we have added one sentence around lines 133-135 where we begin introducing the dataset in detail.
 • It would be ideal to include a sensitivity power analysis given that some effect sizes are very small (e.g., a small monetary difference of contributions between women and men) or don't seem meaningful (e.g., H4). 	We have now included a sensitivity power analysis showing the minimum detectable effects for each coefficient in our main regression in Supplemental Materials A10.
 • The language surrounding the variable of "visibly present females" was confusing. It took me until much later in the paper before I realized that the authors operationalized this by female "past donations" listed on the page and not by somehow being able to tell who was online at the same time as the individual donor. Please make sure this definition is clear as early as possible. 	We have revised the manuscript on lines 625-638 to clarify this.
 • H6 is stated as a mediation model and in a way that cannot be tested by the given data. This is misleading and an exaggeration of what the present data can tell us. 	We have revised the manuscript to clarify that H6 expects that women will leave more empathic messages (lines 401-402, line 417, and lines 669-670).
 • It is misleading how results are presented regarding H3. The effects are not even marginally significant but are interpreted as real effects as first. The authors state only in the end that the results are not significant. 	This is a fair point and we have reworded the section starting at line 608 to make it more immediately clear that these effects are not statistically significant.
 • The authors wrote: "If humans have a predisposition to help family members, we should see significantly higher contributions to recipients who are apparently related to donors than to those who are not. Our second hypothesis, H2, is that a significantly higher amount on average will be donated to recipients who are apparent family members of donors than to those who are not." I found the wording of this passage to be confusing and am unsure how the two predictions differ from each other. 	We thank Reviewer #3 for pointing out that these sentences could be clearer. We have rephrased them in the manuscript (lines 261-266).
 • "Our further analyses of the more than 300,000 self-identified donations test..." I am not sure what "self-identified" means in this context. 	We have revised the manuscript to clarify this throughout. "Self-identified" simply meant that they left their names beside their donations (they were not anonymous donors). We now use the term "non-anonymous" to clarify.

Reviewers' Comments:

Reviewer #2:

Remarks to the Author:

The authors have very thoughtfully and thoroughly addressed my concerns.

I especially appreciate the efforts the authors made to clarify when they were focusing on "ultimate" level explanations, and when they were focusing on the "proximate" psychology.

I thought the results on kin that were included in the authors' response to my comments were fascinating, and while I have no objections to omitting them from this manuscript, I would love it if they see the light of day.

- Erez Yoeli

Reviewer #3:

Remarks to the Author:

These revisions have addressed many of my concerns and improved the paper. I think it would make a solid contribution to Nature Communications.

In particular, the follow-up survey has added precision to the claim about altruistic giving (now 11%, instead of 21%). It's possible there are other selfish factors we didn't consider that may account for part of that 11% and the authors may wish to mention this in the discussion. As such, this might be a liberal estimate in the current sample of anonymous donations.

I am also not fully convinced about some of the causal claims and still think the authors should be more circumspect about this until they have experimental evidence to this effect. But I'll defer to the editor on this issue.

Otherwise, I think the paper would make an interesting contribution to the literature and will likely cite it myself in the future.

Jay Van Bavel

Reviewer #2 (Remarks to the Author):

The authors have very thoughtfully and thoroughly addressed my concerns.

I especially appreciate the efforts the authors made to clarify when they were focusing on "ultimate" level explanations, and when they were focusing on the "proximate" psychology.

I thought the results on kin that were included in the authors' response to my comments were fascinating, and while I have no objections to omitting them from this manuscript, I would love it if they see the light of day.

- Erez Yoeli

Our response:

Regarding the kin findings Reviewer 2 described, we have included mention of them in the discussion section and detailed information about the results in the Supplementary Methods. We feel this is the appropriate place for them in the paper, as opposed to integrated with the main results because this analysis was entirely exploratory without a clear hypothesis based on past literature unlike our main analyses which were based on prior findings.

Reviewer #3 (Remarks to the Author):

These revisions have addressed many of my concerns and improved the paper. I think it would make a solid contribution to Nature Communications.

In particular, the follow-up survey has added precision to the claim about altruistic giving (now 11%, instead of 21%). It's possible there are other selfish factors we didn't consider that may account for part of that 11% and the authors may wish to mention this in the discussion. As such, this might be a liberal estimate in the current sample of anonymous donations.

I am also not fully convinced about some of the causal claims and still think the authors should be more circumspect about this until they have experimental evidence to this effect. But I'll defer to the editor on this issue.

Otherwise, I think the paper would make an interesting contribution to the literature and will likely cite it myself in the future.

Jay Van Bavel

Our response:

We addressed both suggestions of Reviewer 3 in this revision. We note in the discussion (lines 345-346) that there may have been selfish factors we did not consider in the questionnaire although we designed it to be as comprehensive as we could. Additionally in response to Reviewer 3, we point out in the discussion (lines 464-466) that since our data are observational, causal interpretations of our findings must be made cautiously. We have also adjusted the language describing our findings throughout to not make strong causal claims based on our results.